# Neoadjuvant Chemotherapy in Breast Cancer: An Advanced Personalized Multidisciplinary Prehabilitation Model (APMP-M) to Optimize Outcomes

**DOI:** 10.3390/jpm11050324

**Published:** 2021-04-21

**Authors:** Alba Di Leone, Daniela Terribile, Stefano Magno, Alejandro Martin Sanchez, Lorenzo Scardina, Elena Jane Mason, Sabatino D’Archi, Claudia Maggiore, Cristina Rossi, Annalisa Di Micco, Stefania Carnevale, Ida Paris, Fabio Marazzi, Valeria Masiello, Armando Orlandi, Antonella Palazzo, Alessandra Fabi, Riccardo Masetti, Gianluca Franceschini

**Affiliations:** 1Multidisciplinary Breast Centre, Dipartimento Scienze della Salute della Donna e del Bambino e di Sanità Pubblica, Fondazione Policlinico Universitario A. Gemelli IRCCS, 00168 Rome, Italy; daniterribile@gmail.com (D.T.); stefano.magno@policlinicogemelli.it (S.M.); martin.sanchez@hotmail.it (A.M.S.); lorenzoscardina@libero.it (L.S.); elenajanemason@gmail.com (E.J.M.); sabatinodarchi@gmail.com (S.D.); riccardo.masetti@policlinicogemelli.it (R.M.); gianlucafranceschini70@gmail.com (G.F.); 2Centre of Integrative Oncology—Multidisciplinary Breast Centre—Dipartimento Scienze della Salute della Donna e del Bambino e di Sanità Pubblica, Fondazione Policlinico Universitario A. Gemelli I RCCS, 00168 Rome, Italy; claud.maggiore@gmail.com (C.M.); cristina.rossi13@yahoo.it (C.R.); annalisadimicco@nutrimentidimindfulness.it (A.D.M.); 3UOS Psicologia Clinica, Fondazione Policlinico Universitario A. Gemelli IRCCS, 00168 Rome, Italy; dott.stefaniacarnevale@gmail.com; 4Department of Woman and Child Health and Public Health, Woman Health Area, Fondazione Policlinico Universitario A. Gemelli I RCCS, 00168 Rome, Italy; ida.paris@policlinicogemelli.it; 5UOC di Radioterapia Oncologica, Dipartimento di Diagnostica per Immagini, Radioterapia Oncologica ed Ematologia, Fondazione Policlinico Universitario A. Gemelli I RCCS, 00168 Rome, Italy; fabio.marazzi@policlinicogemelli.it (F.M.); valeria.masiello@policlinicogemelli.it (V.M.); 6Comprehensive Cancer Center, Multidisciplinary Breast Unit, Fondazione Policlinico Universitario Agostino Gemelli IRCCS, Largo Agostino Gemelli, 8, 00168 Rome, Italy; armando.orlandi@policlinicogemelli.it (A.O.); antonella.palazzo@policlinicogemelli.it (A.P.); 7Medicina di Precisione in Senologia, Dipartimento Scienze della Salute della Donna e del Bambino e di Sanità Pubblica, Fondazione Policlinico Universitario A. Gemelli IRCCS, 00168 Rome, Italy; alessandra.fabi@policlinicogemelli.it

**Keywords:** breast cancer, neoadjuvant chemotherapy, multidisciplinary treatment, evidence-based medicine, personalized treatment, oncological outcomes, patient quality of life

## Abstract

Neoadjuvant chemotherapy is increasingly being employed in the management of breast cancer patients. Efforts and resources have been devoted over the years to the search for an optimal strategy that can improve outcomes in the neoadjuvant setting. Today, a multidisciplinary approach with the application of evidence-based medicine is considered the gold standard for the improvement of oncological results and patient satisfaction. However, several clinical complications and psychological issues due to various factors can arise during neoadjuvant therapy and undermine outcomes. To ensure that health care needs are adequately addressed, clinicians must consider that women with breast cancer have a high risk of developing “unmet needs” during treatment, and often require a clinical intervention or additional care resources to limit possible complications and psychological issues that can occur during neoadjuvant treatment. This work describes a multidisciplinary model developed at “Fondazione Policlinico Universitario Agostino Gemelli” (FPG) in Rome in an effort to optimize treatment, ease the application of evidence-based medicine, and improve patient quality of life in the neoadjuvant setting. In developing our model, our main goal was to adequately meet patient needs while preventing high levels of distress.

## 1. Introduction

Breast cancer patients that exhibit high tumor-to-breast volume ratio, lymph node-positive disease, and aggressive biological features (high grade, hormone receptor-negative, HER2-positive, triple negative characterization) are more often candidates for neoadjuvant chemotherapy (NAC). Although large clinical trials have shown no differences in terms of overall and disease-free survival between adjuvant and neoadjuvant systemic therapy, NAC may provide important advantages [1,2]: tumor chemosensitivity can be assessed in vivo by monitoring the response to therapy, potentially allowing for the switching of therapies in case of non-responsiveness; downstaging of tumors often allows clinicians to favor breast-conserving surgery (BCS) over mastectomy and contain excision volumes, thus improving cosmetic results; downstaging of the axilla can allow for the avoidance of lymph node dissection in selected patients, reducing surgical morbidity [3] (Figure 1). Therapeutic regimens include anthracyclines (epirubicin, 100 mg/m^2^), cyclophosphamide (500 mg/m^2^; triweekly for 4 cycles) and taxanes (docetaxel, 70 mg/m^2^; triweekly for 4 cycles); or carboplatin (100 mg/m^2^; weekly for 12 cycles); taxanes are combined with targeted trastuzumab therapy in case of HER2-positivity.

Specific evidence-based guidelines have been released to ensure that each patient treated in the neoadjuvant setting may receive the most effective, evidence-based chemotherapy regimen, in a personalized, multidisciplinary setting (Figure 2).

Less attention has been devoted to addressing the specific “unmet needs” that patients may experience during treatment [5]. The benefits of a multimodal prehabilitation model are still emerging in recent studies, particularly during the preoperative period. During this window of opportunity, patients may be more receptive to health behavior changes in a structured support network [6].

In this paper, we present the details of an advanced, personalized, multidisciplinary prehabilitation protocol, which we have adopted in our Multidisciplinary Breast Center at Fondazione Policlinico Universitario Agostino Gemelli (FPG) in Rome since 1 May 2018 for patients scheduled to receive NAC.

This protocol allows patients to access not only the most appropriate, evidence-based chemotherapy regimen, but also specific interventions aimed at protecting their quality of life via the inclusion of lifestyle and nutrition counselling, along with psychological distress- and integrative oncology (IO)-complementary interventions [7].

## 2. Materials and Methods

Our breast unit treats approximately 1000 new breast cancer cases every year. Between 1 May 2018 and 31 December 2020, 250 patients were referred to our center for neoadjuvant treatment. The mean patient age was 53 (range 25–74), and 130 patients were premenopausal.

Our broad-based interdisciplinary team includes ten breast surgeons, two medical oncologists, two breast pathologists, five breast radiologists, three breast radiologic technicians, three psycho-oncologists, two nutritionists, two integrative oncology physicians, six certified breast care nurses, and one data manager, all exclusively devoted to the management of patients with breast disease. Other team members that devote at least 50% of their activity to breast pathology include three plastic surgeons, three additional medical oncologists, two radiation oncologists, two oncogeriatricians, two gynecologists, one geneticist, one cardiologist, and two palliative care specialists. All specialists regularly attend weekly multidisciplinary meetings (MDMs), in which all new cases of breast cancer are discussed [8]. In this setting, patients are also evaluated for enrollment in clinical trials [9].

During MDMs, the case of every patient is discussed in detail, and an individualized treatment plan is programmed in adherence to the latest practice guidelines. Out of 250 patients, 98 were scheduled for an appointment with the geneticist, 14 were referred for fertility counseling, and an appointment with a geriatrician was arranged for 34 elderly patients.

## 3. Breast Unit and Outpatient Neoadjuvant Care Prehabilitation Clinic

All patients receiving an indication to NAC are referred to the “outpatient neoadjuvant care prehabilitation clinic”, where they are jointly taken care of by a “neoadjuvant oncologic treatment team” and a “neoadjuvant supportive care team”. The first team explains the care plan designed by the multidisciplinary panel, and brings into focus the important aspects of their respective areas of expertise. At the end of the interview, the patients are directed to a follow-up examination by the supportive care team for a complete psychological, nutritional, and lifestyle assessment that will serve as a baseline for the upcoming treatment. As a result, the treatment is tailored to every patient in a multidisciplinary, holistic fashion [10]. When possible, every appointment is scheduled on the same day, to limit patient discomfort in returning to the hospital several times in the same week.

## 4. The Neoadjuvant Oncologic Treatment Team

In this setting, patients are welcomed by a team of experts consisting of a breast surgeon together with the patient’s referring oncologist and breast nurse (Figure 3) [11].

This treatment team is in charge of reviewing the diagnostic workup, discussing the therapeutic plan, and explaining, scheduling, and monitoring additional interventions that may be relevant according to the age and specific medical features of each individual patient.

As a first step, the oncologic team reviews the diagnostic workup and schedules any additional appointments that may be required to complete it.

Every patient undergoing NAC in our breast center must have completed a full diagnostic panel that includes [12]:Clinical breast examination, mammography, breast ultrasound, and breast MRI;Ultrasound- or stereotactic-guided tissue sampling of breast lesions and suspicious lymph nodes. Markers are positioned in the breast tissue and pathologic lymph nodes in order to ensure a correct pre-surgical localization in case of pathologic complete response or regression to a non-palpable lesion;Complete histopathological and prognostic characterization (ER, PgR, AR, Ki67, HER2 status);Photographical documentation of pre-NAC patient breasts. After clinical and ultrasound evaluation, the surgeon draws the tumor’s projection and measurements on the skin surface and takes two photographs in frontal and lateral projection (Figure 4). Pictures are re-evaluated after NAC and assist in surgical planning [13];Systemic staging is completed by performing either a whole-body CT scan and bone scintigraphy, or a PET/CT scan.

The team then reviews with the patient the global therapeutic plan [14]. The oncologist discusses with the patient the details of the chemotherapeutic regimen (previously defined at the MDM) and a date for the first session of NAC is set. An appointment for central venous catheter placement is also provided.

Based on age, general conditions, family history, and pathologic features of the tumor, the following additional interventions are discussed and eventually scheduled.

### 4.1. Cardiovascular Assessment

As conventional chemotherapy and targeted therapies are associated with an increased risk of cardiac damage [15], each patient scheduled for NAC undergoes preliminary cardiovascular assessment.

The development of cardiotoxicity, even if asymptomatic, not only adversely affects patient cardiac prognosis, but may significantly limit the proper completion of therapeutic protocols, especially if additional anticancer treatments become necessary after recovery/relapse of the disease [16]. Cardiovascular disease is now the second leading cause of long-term morbidity and mortality among cancer survivors, and the leading cause of death among female breast cancer survivors [17].

Our protocol ensures that a cardio-oncologist evaluates the patient via electrocardiogram and echocardiography before beginning treatment, and then periodically in relation to personal risk and ongoing pharmacological treatment. An adequate preliminary stratification of cardiotoxicity risk and the early identification and treatment of subclinical cardiac damage may help to avoid withdrawal of chemotherapy and prevent irreversible cardiovascular dysfunction.

### 4.2. Genetic Counselling

Because of recent media and popular culture coverage, general knowledge about breast cancer genetics has increased in recent years [18]. Genetic test results have also become increasingly relevant in selecting the most effective systemic therapy, thanks to the advent of PARP inhibitors for treatment of BRCA1/2-associated breast cancers. Genetic assessment has become equally relevant for the optimization of radiation therapy, with emerging concerns about radiation safety for the carriers of certain pathogenic mutations (e.g., TP53) [19].

In our model, indications to genetic testing are discussed for every patient during the MDM, taking into account patient age, family history, and the clinical features of the disease. If, according to current Italian and American guidelines [20,21], genetic testing is considered appropriate, an interview with the clinical geneticist is immediately scheduled [22]. The advantage of this approach is that patients who then undergo NAC have approximately six months to complete a full, well-rounded genetic evaluation before the scheduling of surgery. This allows us to tailor surgical choices based on the test results, avoiding the unnecessary double surgery that could derive from a positive test result obtained after breast-conserving surgery [23].

### 4.3. Multiparametric Geriatric Assessment in Elderly Patients

Elderly patients represent a very heterogeneous community in terms of life expectancy, comorbidities, and cognitive and social function, therefore it is crucial not to deny treatment based on age alone. In this framework, a multiparametric geriatric assessment is always appropriate, and is a convenient supplement in the evaluation of every elderly patient treated for breast cancer, as it can move the needle on proposed treatment.

A recent study by Okonji et al. reported that nearly 50% of fit elderly women with high-risk disease are undertreated [24]. The neoadjuvant use of chemotherapy is further neglected, with studies reporting higher toxicity rates and lower incidence of complete pathological response in patients aged over 65 [25]. However, although elderly patients are generally underrepresented in clinical trials, those with non-triple negative breast cancer show a prognosis comparable to younger patients in terms of overall survival [26]. An individualized care model must therefore be applied to select the elderly sub-population that could benefit from neoadjuvant chemotherapy, and monitor it closely during treatment to prevent toxicity in these fragile patients [27].

In our protocol, patients aged 70 years or older, whether or not they exhibit relevant comorbidities, are scheduled for a pre-treatment comprehensive geriatric assessment. The assessment is performed by a dedicated geriatrician with experience in breast oncology, who actively participates in our multidisciplinary team. Comorbidities, cognitive and psychological disorders, physical performance, risk factors, nutritional status, and general autonomy are comprehensively evaluated, and NAC is scheduled only in the event of oncogeriatric clearance. A second assessment is also scheduled at the end of chemotherapy.

### 4.4. Gynecologic and Fertility Counselling in Younger Patients

Chemotherapy and/or ovarian suppression can cause early (permanent or temporary) menopausal symptoms and reduce fertility. Many women are concerned about these issues, and it is important to provide them with proper counseling and treatment [28]. As regards menopausal symptoms, a large number of patients find these difficult to cope with, with a significant negative impact on their quality of life. Our gynecologists manage these symptoms using both traditional medicine and integrative care.

Fertility care should follow a multidisciplinary team-based approach, with strict interaction between medical oncologists, surgeons, and fertility specialists [29,30]. In our multidisciplinary prehabilitation care model, the main goal is to preserve the opportunity for family planning, offering oncofertility services in a timely manner without delaying chemotherapy.

The breast nurse follows the patient on this pathway and during the subsequent procedures for ovarian function and/or fertility preservation.

## 5. The Neoadjuvant Supportive Care Team

After completing the assessment with the “oncological treatment team”, every patient is directed to a meeting with the “neoadjuvant supportive care team”, which includes a nutritionist, a psycho-oncologist, and an integrative oncology expert (Figure 5) [31].

A lifestyle interview is conducted, and anthropometric parameters and body composition analysis are measured via segmental multi frequency–bioelectrical impedance analysis (SMF-BIA).

Nutritional and physical activity screenings are performed in our unit just before the beginning and at the conclusion of oncologic treatments, and the impact of each type of intervention, from surgery to chemotherapy, on BMI, body composition, and metabolism is monitored during therapy. In this regard, patients are asked to keep a diary and send it regularly via email, and periodic video interviews are scheduled [32,33,34].

### 5.1. Lifestyle and Nutrition Counseling

Physical activity (PA), nutrition, body weight, and metabolism all play a key role in almost every aspect of cancer onset, progression, and management [35] (WCRF -World Cancer Research Fund 2018). However, nutritional screening is seldom performed even in high-quality breast units, and data on its value are still scarce [36,37,38].

Specific recommendations about diet and physical activity based on the most recent scientific evidence [35] are given to all patients, with the aim of relieving chemotherapy toxicity and improving quality of life and oncological outcomes [39,40]. Moreover, during and after treatments, patients are supported by a personalized nutritional approach and motivated to practice PA in order to decrease their disease recurrence risk [39,41]. PA during cancer treatments represents a powerful asset to improve therapy-induced conditions such as anxiety, depression, sleep disorders, lymphedema, cancer- and therapy-related fatigue, bone health, and overall quality of life [42,43,44,45,46].

### 5.2. Psychological Counselling

Chemotherapy generates a distress that, over time, can severely affect patient quality of life [47,48]. A recent study showed that post-NAC patients have a significantly higher level of distress compared to patients receiving chemotherapy after surgery [49]. Understanding the needs of patients undergoing NAC enables us to address the communication process more appropriately, provide psychological support, and build clinical and rehabilitation interventions in a more personalized way [47]. In line with NCCN guidelines, the diagnostic and therapeutic pathways of patients scheduled for NAC include a pre–post treatment psychological evaluation. Specific, psycho-oncological support should be given to patients undergoing chemotherapy. At the beginning of NAC, patients undergo a clinical psychological interview aimed at assessing their risk of oncological distress, and identifying both the dysfunctional psychological factors and the protective psychosocial factors that could affect treatment. The goal is to improve adaptation to the oncological disease and promote adherence to therapeutic treatments. In addition to the interview, a psychometric assessment is carried out through screening and the employment of clinical tools such as the Distress Thermometer (DT) [50], the Hospital Anxiety Depression Scale (HADS) [51] and the General Self-Efficacy Scale (GSES) [52].

In our breast center, we aim to validate a semi-structural interview, which leads to a holistic and trans-disciplinary measurement of the psychological state of the patients. The assessment allows us to identify patients who may benefit from a psychological support intervention, individual psychotherapy, or group therapy [53] (Figure 6).

#### Emotional Eating Prevention during NAC

The impact of psychosocial factors such as worry, perceived risk, and perceived treatment efficacy on diet has been understudied in breast cancer patients [54]. The relationship between distress, weight change, and nutrition has been the subject of a fairly recent psycho-oncological study trend, with major studies conducted on patients at the end of their therapies. Our model proposes an integrative approach to identify emotional eating, a dietary pattern wherein people use food to help them deal with stressful situations and in response to negative emotions. Overweight individuals have been found to exhibit less effective coping skills in response to negative emotions, which leads them to emotionally eat more frequently [55]. Psychological disciplines can help to identify healthy and harmful habits, and promote changes in attitudes and healthy behaviors.

Psychological intervention based on the activation of self-efficacy in dietary behavior could favor the ability to adapt to oncological therapies through active participation in treatment, redefinition of problems, and the evaluation of alternative solutions. At the same time, the intervention acts in support of lifestyle changes and involves the activation of specific psychoeducational groups for patients who need to change their dietary behavior.

### 5.3. Integrative Oncology during Neoadjuvant Therapy

Our patients receive information about evidence-based complementary therapies available, in order to optimize the management of symptoms related either to the disease itself or to treatment toxicity: most frequently gastro-intestinal disorders, hot flashes, fatigue, insomnia, mucositis, peripheral neuropathy, anxiety, and mood disorders.

In accordance with the SIO (Society of Integrative Oncology) clinical guidelines for breast cancer patients [56] recently endorsed by the American Society of Clinical Oncology (ASCO) [7], personalized integrative care plans at the FPG Center for Integrative Oncology include mind–body interventions such as acupuncture, mindfulness-based protocols, qi gong, massage therapy, and other group programs like music therapy, art therapy, and therapeutic writing workshops.

#### 5.3.1. Acupuncture

Acupuncture, well known as a branch of traditional Chinese medicine, represents a reliable, cost-effective, and safe procedure for symptom management, if performed properly and by a specialized practitioner. The NCCN recommends acupuncture for pain, fatigue, nausea/vomiting, and hot flashes [57]. For some of these symptoms, such as nausea/vomiting, acupuncture can be used as a valid option for patients who wish to avoid pharmacological treatment. For other conditions, including fatigue, hot flashes, and chemotherapy-induced peripheral neuropathy (CIPN), acupuncture should be considered when conventional treatments are ineffective, not available, or burdened by remarkable side effects.

In many patients experiencing hot flashes due to chemotherapy-induced ovarian failure and/or estrogen-blocking treatments, a course of 6–12 acupuncture treatments is associated with therapeutic effects that persist for six months or longer and do not appear to require prolonged treatment [58,59].

Chemotherapy-induced peripheral neuropathy is a challenging pain symptom to manage, and has been a hot topic for acupuncturists for a long time. A recent Cochrane review concluded that there was insufficient evidence to support or reject the use of acupuncture for neuropathic pain [60]. To date, some randomized phase II studies on the effects of acupuncture are very promising [61,62,63]. Recently published ESMO (European Society of Medical Oncology) guidelines on therapy-induced neurotoxicity [64] state that “Acupuncture might be considered in selected patients to treat CIPN symptoms “(grade II, C). Currently, our Center for Integrative Oncology is taking part in a multicenter clinical trial on acupuncture for chemotherapy-induced peripheral neuropathy (CIPN) in breast cancer.

#### 5.3.2. Mindfulness

Mindfulness is defined as present-moment nonjudgmental awareness, and its practice can take the form of formal meditation, or more informal practices, such as simply remembering to be present as one undertakes day-to-day tasks. Mindfulness-based stress reduction (MBSR) has been shown to reduce distress and improve psychological well-being in patients with cancer [65,66,67]. Preliminary evidence suggests that MBSR may produce effects comparable to pharmacologic treatment for primary insomnia [68] and positively impact sleep quality and quantity in patients with cancer [69,70,71]. Randomized trials of mindfulness-based stress reduction report decreased fatigue, depression, anxiety, and fear of recurrence [72,73].

In addition, improvements have been noted in sleep [72,74], quality of life, and psychosocial adjustments [70], as well as in the long-term adverse effects associated with treatment [65].

## 6. Conclusions

Specific clinical complications and psychological issues due to the disease and therapies can occur during the course of neoadjuvant therapy, undermining outcomes.

We have therefore developed a multidisciplinary model to ease the application of evidence-based oncologic protocols, ensure patient-centered optimal treatment, prevent distress, and improve patient quality of life. Our model of intervention can encourage clinicians to personalize supportive care medicine and direct it towards precision medicine. The development of an appropriate clinical pathway, with multidisciplinary competence and the performance of standardized tasks, is essential in order to obtain a successful treatment and make the patient co-responsible for optimized results within the neoadjuvant setting. However, multidisciplinary prehabilitation trials in breast cancer patients undertaking NAC are necessary to confirm the efficacy of this model.

## Figures and Tables

**Figure 1 jpm-11-00324-f001:**
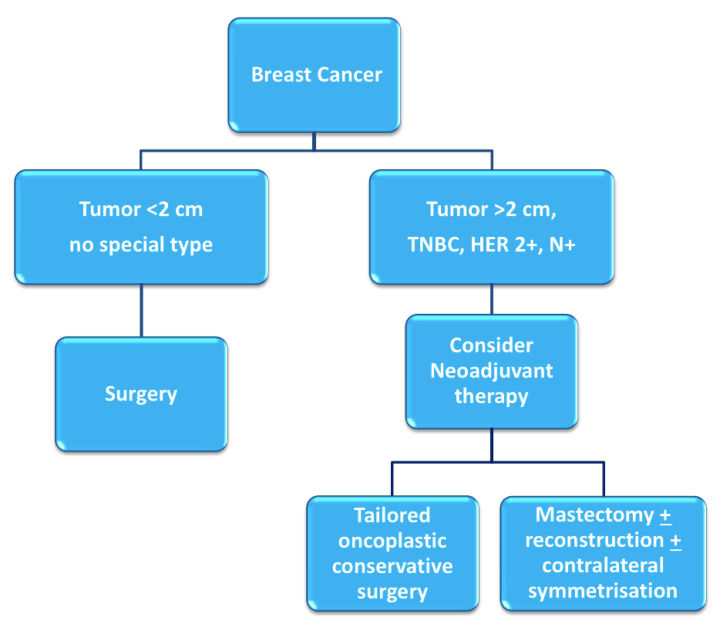
Decision-making process for neoadjuvant chemotherapy [4].

**Figure 2 jpm-11-00324-f002:**
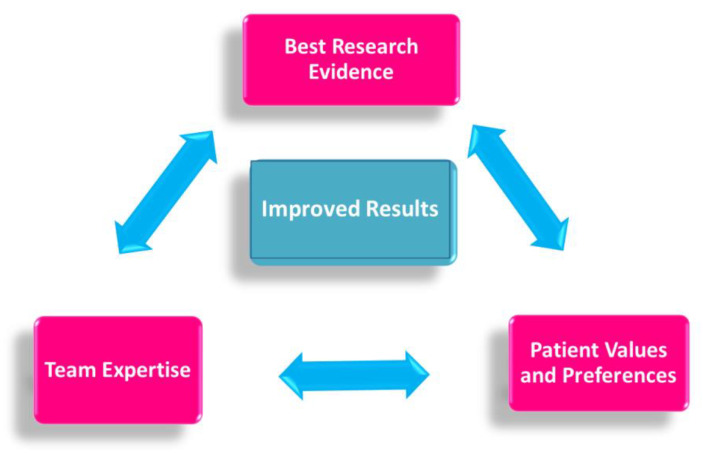
Evidence-based medicine in neoadjuvant chemotherapy.

**Figure 3 jpm-11-00324-f003:**
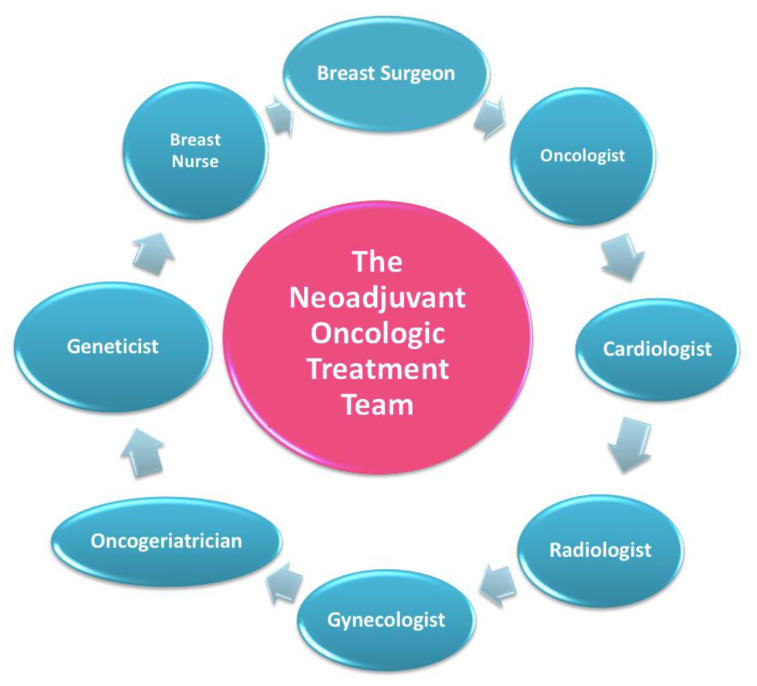
The neoadjuvant oncologic treatment team.

**Figure 4 jpm-11-00324-f004:**
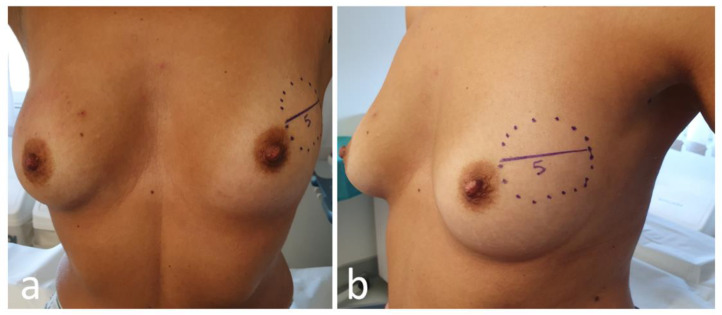
Frontal (**a**) and lateral (**b**) view of pre-neoadjuvant chemotherapy (NAC) breast with tumor projection and measurement (cm).

**Figure 5 jpm-11-00324-f005:**
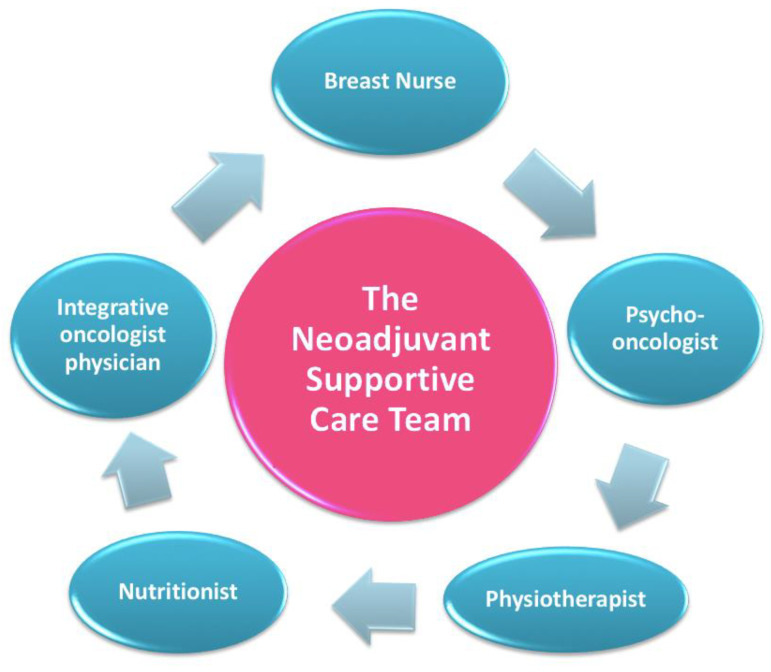
The neoadjuvant supportive care team.

**Figure 6 jpm-11-00324-f006:**
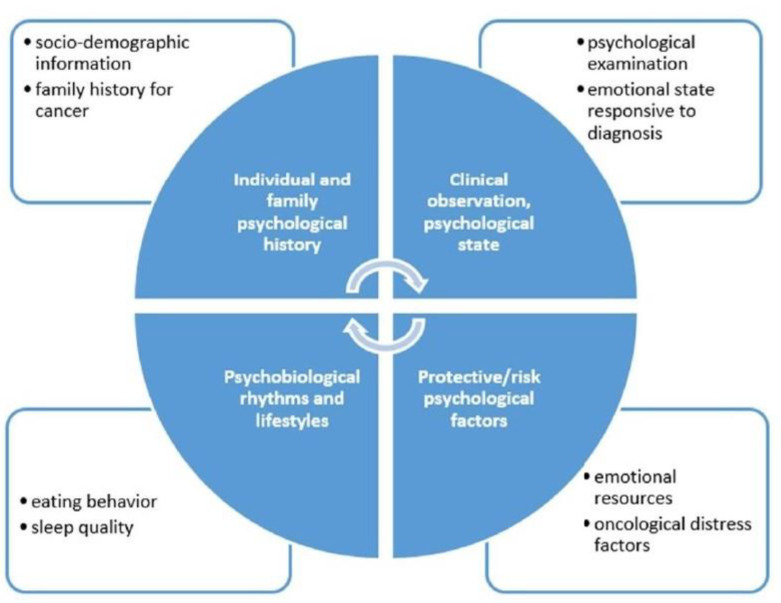
Psychological interview.

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
