# Peer review of "Neoadjuvant Chemotherapy in Breast Cancer: An Advanced Personalized Multidisciplinary Prehabilitation Model (APMP-M) to Optimize Outcomes"

_jpm, 2021, doi:10.3390/jpm11050324_

Round 1

Reviewer 1 Report

The introduction of a chapter called "Materials and Methods" is mandatory, in my opinion in order to verify the effectivenes of the proposed management model. Every feature presented in the article is normally part of a current Breast Unit, so the Authors wouldn't talk about somewhat new if they hadn't brought their own experience.

The proposed method may be interesting but I think some note about methods and results should be reported.

Definitely, there are too many self-citations among references.

Author Response

Dear reviewer,

Thank you for your comments. We understand your concerns, and have revised our manuscript sensibly following your suggestions in order to make our contribution more significant. As you correctly pointed out, a “Materials and Methods” section adds structure to our work, and we took the opportunity to share some preliminary data which has been added to this paragraph. However, we did not add a “results” section: this paper was not intended as a clinical trial, and we believe that adding too much data would lengthen the paper and drive focus away from our main objective, which was to propose an integrative management model that could take care of many still underrepresented aspects of the breast cancer patient in the neoadjuvant setting. We are, however, collecting data in view of a new project which will focus exclusively on clinical results.

We believe our model enhances the traditional “Breast Unit” management you mentioned, adding precious enrichments (such as routine oncofertility screening, mindfulness, acupuncture etc) that, while not formally necessary in a strictly clinical setting, certainly add value to treatment in terms of patient unmet needs and quality of life. To further stress the difference between our proposed model and a traditional Breast Unit management, we added several specifications throughout the text, particularly in the “integrative oncology” section.

We apologize if you felt that self-citations were too many.

However, following your suggestion, we have cut our self-citations to the strictly necessary. We also detected and amended a few linguistic inaccuracies throughout the text, and generally polished the English.

Reviewer 2 Report

I have few concerns:

  1. Please add a brief section in the 'Introduction' describing Neoadjuvant chemotherapy in general (What is this therapy, current applications in breast cancer and others, efficacy compared to adjuvant chemotherapy).
  2. Patient's accessibility to this program is not clearly mentioned in this review. Is this program available for only local patients or international patients as well? Is remote consultation available? It will be great to discuss these points.
  3. Please recheck the manuscripts for typos (example - Line 71 - 1.000 instead of 1,000)
  4. Fig 3 and 5 are low quality/resolution. It will be great if both of them can be replaced with high res images. 

Author Response

Thank you for your review, as it has given us the opportunity to correct and enrich our paper. We have edited our work by means of your suggestions as follows:

  1. We added some general specifications about neoadjuvant chemotherapy in the introduction, and stressed that, while its efficacy in terms of overall survival compared to adjuvant chemotherapy is still debated, neoadjuvant treatment adds some important advantages. We also added some details about treatment regimens.
  2. At the moment our model is fully operative only at our Integrative Breast Centre in Policlinico Gemelli, Rome. While hardly entirely applicable via remote interviews, we nonetheless believe this model is fairly reproducible in every Breast Centre, provided you add the necessary specialists to the team. This is why our current paper is focused on explaining in detail every step of our patient management. Taking insight from your consideration, we took the opportunity to add several specifications throughout the text, including a proper “materials and methods” section, to better explain the course of our management.
  3. We rechecked the whole text for typos and spelling mistakes, and generally refurbished the English.
  4. Figures 3 and 5 are now high quality

Round 2

Reviewer 1 Report

I agree with your changes